# Differences in the Expression Patterns of *TGFβ* Isoforms and Associated Genes in Astrocytic Brain Tumors

**DOI:** 10.3390/cancers14081876

**Published:** 2022-04-08

**Authors:** Natalia Kurowska, Barbara Strzalka-Mrozik, Marcel Madej, Klaudia Pająk, Celina Kruszniewska-Rajs, Wojciech Kaspera, Joanna Magdalena Gola

**Affiliations:** 1Department of Molecular Biology, Faculty of Pharmaceutical Sciences in Sosnowiec, Medical University of Silesia, 40-055 Katowice, Poland; natalka.kurowska@gmail.com (N.K.); mmarcel281297@gmail.com (M.M.); biolmolfarm@sum.edu.pl (K.P.); ckruszniewska@sum.edu.pl (C.K.-R.); jgola@sum.edu.pl (J.M.G.); 2Department of Neurosurgery, Regional Hospital, Medical University of Silesia, 41-200 Sosnowiec, Poland; wkaspera@sum.edu.pl

**Keywords:** brain, tumor, *TGFβ* mRNA isoforms, microarray, cancer, expression pattern

## Abstract

**Simple Summary:**

Numerous molecular changes are observed during tumor progression. Genes associated with TGFβ isoforms are involved in many cancers, including brain cancer. Using molecular techniques to evaluate 43 brain tumor sections from patients at different stages of astrocytic brain tumor, we assessed differences in the expression patterns of genes associated with TGFβ isoforms and quantified the mRNA of three *TGFβ* isoforms. Our study confirmed significant differences in the expression of genes associated with TGFβ isoforms as well as differences in isoform expression and the identification of 16 genes that differentiate between disease grades. Database analysis revealed interactions between the products of these genes, some of which are associated with tumorigenesis. Differences in the expression patterns of transcripts associated with TGFβ isoforms confirm that they are involved in astrocytic brain tumor transformation. Quantitative assessment of *TGFβ2* mRNA may be a useful method in the future to facilitate the diagnosis of disease grade.

**Abstract:**

Genes associated with the TGFβ isoforms are involved in a number of different cancers, and their effect on the progression of brain tumors is also being discussed. Using an oligonucleotide microarray method, we assessed differences in expression patterns of genes in astrocytic brain tumor sections from 43 patients at different stages of disease. Quantitative mRNA assessment of the three *TGFβ* isoforms was also performed by real-time RT-qPCR. Oligonucleotide microarray data were analyzed using the PL-Grid Infrastructure. The microarray analysis showed a statistically significant (*p* < 0.05) increase in *TGFβ1* and *TGFβ2* expression in G3/G4 stage relative to G2, whereas real-time RT-qPCR validation confirmed this change only for the *TGFβ2* isoform (*p* < 0.05). The oligonucleotide microarray method allowed the identification of 16 differential genes associated with *TGFβ* isoforms. Analysis of the STRING database showed that the proteins encoded by the analyzed genes form a strong interaction network (*p* < 0.001), and a significant number of proteins are involved in carcinogenesis. Differences in expression patterns of transcripts associated with *TGFβ* isoforms confirm that they play a role in astrocytic brain tumor transformation. Quantitative assessment of *TGFβ2* mRNA may be a valuable method to complement the diagnostic process in the future.

## 1. Introduction

Gliomas, which include astrocytomas (i.e., tumors arising from astrocytes), are one of the most frequent brain tumors and are classified into four grades depending on their characteristics and severity. Glioblastoma multiforme (GBM), classified as a grade IV astrocytoma, is a prevalent and malignant tumor [1,2]. The average survival time among patients is less than 15 months with standard therapies [3,4].

Achievements in molecular biology have led to a new classification of brain tumors approved by the World Health Organization (WHO) in 2016 based on genome characterization and the identification of epigenetic changes [5]. The latest version of the Central Nervous System Tumors Classification, in use since 2021, particularly emphasizes the growing importance of molecular biology methods in the diagnosis and classification of brain tumors. Molecular biomarkers provide both defining and supporting diagnostic information. Some tumors can be uniquely identified on the basis of molecular alterations, while for others this provides only ancillary information or does not facilitate their classification. As an example, a tumor may be diagnosed as an astrocytoma, IDH mutant after characteristic molecular changes have been found in the *IDH1*, *IDH2*, *ATRX*, *TP53*, *CDKN2A/B* genes. Determination of the methylation profile of tumors is also of great importance in the classification of brain tumors, which allows accurate identification of almost all brain tumors and is currently used auxiliary in diagnosis in combination with traditional histological methods. The combination of traditional histological and molecular methods is also expected to facilitate prognostic estimation in brain tumors, as it is acceptable to classify a tumor as malignant when characteristic molecular alterations are present, even if histological examination suggests a lower degree of malignancy [6]. Undoubtedly, the new WHO Classification of Tumors of the Central Nervous System indicates an increasing role for molecular diagnostics in the future of neurooncology. 

Cancer progression is associated with a multitude of molecular and immunological changes, and the same is true for brain tumors. Many reports indicate a dysfunction of the production of various cytokines in glioblastoma, including transforming growth factor β (TGFβ), which plays a key role in its development [4]. 

TGFβ has six isoforms, only three of which have been described in humans: TGFβ1, TGFβ2, and TGFβ3 [7]. These isoforms show almost 71–79% amino acid identities but are encoded by three different genes [8]. Their biological activity depends on the quantitative relationships between the different isoforms [9]. 

TGFβ family plays an important role in both normal and pathological cells. These cytokines regulate such fundamental aspects of cellular function as cell growth, differentiation, inflammation and wound healing [9].

TGFβ belongs to anti-inflammatory cytokines and is secreted by immune cells after injury [10]. Cekanaviciute et al. [11] demonstrated that in response to *Toxoplasma gondii* infection, the TGFβ signaling pathway was activated in mouse astrocytes, which plays an important role in astrocytes controlling the neuroinflammatory response during infection. In turn, inhibition of TGFβ signaling is associated with increased infiltration of immune cells in response to the pathogen, increased levels of pro-inflammatory chemokines and cytokines and neuronal damage. 

TGFβ also has other functions in the nervous system. TGFβ1 is the most widely expressed isoform and is associated with injury and is involved in astrocyte scar formation in response to brain injury [12]. In astrocyte culture, TGFβ enhances the expression of neurocan, a chondroitin sulphate proteoglycan that mediates glial scar formation. Furthermore, TGFβ1 injection induces a scarring response [10]. TGFβ1 may also be a key regulator of astrocyte survival and differentiation, angiogenesis, brain homeostasis, memory formation and neuronal plasticity [10,12]. Moreover, this cytokine regulates the phenotype of glial cells, as it has been shown that it can inhibit the mitotic effects of fibroblast growth factor and epithelial growth factor on astrocytes and suppress their proliferation. TGFβ1 may also act as a chemotactic factor for astrocytes in a dose-dependent manner [10].

Physiologically, the TGFβ family has many other functions, ranging from the maintenance of the body’s homeostasis and participation in tissue embryogenesis to the activation of cellular cytostatic and cell death processes [13]. Paradoxically, this cytokine inhibits cell proliferation and stimulates differentiation, thus acting as a tumor suppressor. It is responsible for the activation of apoptosis and autophagy, the suppression of angiogenesis, and the inhibition of inflammatory processes [14,15,16]. However, in advanced cancer, TGFβ induces tumor progression and metastasis, acting as an oncogenic factor. It also stimulates extracellular matrix remodeling, promotes angiogenesis, and silences the immune cell response by affecting the tumor microenvironment [16]. Another important role of TGFβ is its involvement in epithelial–mesenchymal transition [17]. “The TGFβ paradox” is reflected in the clinic, where in the early stages of cancer, TGFβ levels are positively correlated with prognosis. In advanced tumors, the level of this cytokine correlates with tumor size, invasiveness, and atypia, thus potentially becoming a useful biomarker in the future [15,16,18].

Genes dependent on the TGFβ pathway play an important role in lung, pancreatic, breast, colorectal, and melanoma cancers [19,20,21,22]. The effect of this cytokine on the development of gliomas has also been studied [23,24,25,26]. However, the quantitative relations between *TGFβ1*, *TGFβ2*, and *TGFβ3* isoforms and associated genes in astrocytic brain tumors remain unclear.

Therefore, the aim of our study was to assess differences in the expression patterns of genes associated with *TGFβ* isoforms in astrocytomas with respect to the degree of malignancy. We also quantified the mRNA of the three *TGFβ* isoforms—*TGFβ1*, *TGFβ2,* and *TGFβ3*—which may prove helpful in the diagnostic process and facilitate the correct determination of the extent of the lesion.

## 2. Materials and Methods

### 2.1. Subjects

Specimens of astrocytic sections of brain tumors were collected from 43 patients with a mean age of 54 ± 14 years. The diagnosis of a brain tumor, qualification for surgery, and resection of the lesion were performed at the Department and Clinical Department of Neurosurgery of the Medical University of Silesia in the Provincial Hospital of St. Barbara in Sosnowiec, Poland.

The initial diagnosis of brain glioma in the patients was established by contrast-enhanced computed tomography. This was followed by magnetic resonance imaging (MRI) using T1- and T2-weighted sequences, fluid-attenuated inversion recovery sequence (which is a modification of the T2-weighted sequence), and/or diffusion tensor imaging. In the case of tumor localization near the eloquent areas of the brain, the above-mentioned basic MRI sequences were extended by functional MRI and diffusion MRI tractography to use these sequences in the neuronavigation system. 

During surgery, surgical resection was performed as widely as possible using neuronavigation, fluorescence imaging using gliolate five-aminolevulinic acid, and, for tumors located near the sensorimotor cortex, electrical stimulation of the brain. 

The final diagnosis was confirmed by histopathological evaluation of the resected lesion. Based on the histopathological examination, tumor malignancy grades were determined according to the WHO scale: grade II (G2), grade III (G3), and grade IV (G4) (Table 1).

Patient eligibility for the study was based on the inclusion and exclusion criteria. The inclusion criteria were as follows: patients hospitalized in the Department of Clinical Neurosurgery at the Medical University of Silesia who were qualified for surgery for an astrocytic section brain tumor and patients who gave informed consent to participate in the study. Patients who had used angiotensin-converting enzyme inhibitors for at least three months and those with NYHA class III and IV cardiovascular failure, renal failure, or another neoplastic lesion were not included in the study.

### 2.2. Tissues

All analyzed tissues were divided into 2 parts: one was examined pathomorphological, and the other was immediately stored in RNAlater^®^ (Sigma-Aldrich, St. Louis, MO, USA) at −20°C for 24 h until RNA extraction. 

The research was approved by the local Bioethics Committee.

### 2.3. Total RNA Extraction

Fragments of brain tumors of the astrocytic sections were homogenized using a Polytron^®^ homogenizer (Kinematics AG, Uster, Switzerland). After that, total RNA was extracted with the use of TRI Reagent (Sigma-Aldrich, St. Louis, MO, USA) according to the manufacturer’s protocol. Purification was performed using the RNeasy Mini Kit (Qiagen Inc., Hilden, Germany, Cat. No. 74106) using columns and DNase I (RNase-Free DNase Set, Qiagen Inc., Hilden, Germany, Cat. No. 79254).

The extracted total RNA was qualitatively and quantitatively evaluated. In the qualitative assessment, the technique of electrophoresis in 1% agarose gel using SimplySafe (EurX, Gdansk, Poland) as a dye was used. Quantitative evaluation of the isolated total RNA was performed by spectrophotometric measurement using a MaestroNano MN-913 nanospectrophotometer (MaestroGen Inc., Las Vegas, NV, USA). The extracted material was found to be of normal purity.

### 2.4. Oligonucleotide Microarray Analysis

Ten samples of brain tumor sections were selected for pre-analysis on oligonucleotide microarrays (4 samples; grade II [G2], and 6 samples grade III and IV [G3/G4]). The gene expression profile study was based on an oligonucleotide microarray method using HG-U133A 2.0 plates from Affymetrix (Santa Clara, CA, USA) according to the manufacturer’s recommendations. Synthesis of cDNA was performed with the use of SuperScript^®^ Choice System kit (Invitrogen Life Technologies, Waltham, MA, USA). Biotinylated cRNA was synthesized from the cDNA array using the BioArray HighYield RNA Transcript Labelling Kit (Enzo Life Sciences, Inc, Farmingdale, NY, USA) as the next step of analysis. After then the fragmentation of biotin-labeled cRNA was performed by the Sample Cleanup Module kit (Qiagen GmbH, Hilden, Germany) for 35 min at 94 °C. Obtained cRNA was hybridized to the HG-U133A microarray, labeled with phycoerythrin-streptavidin complex and scanned using GeneArray Scanner G2500A (Agilent Technologies, Inc., Santa Clara, CA, USA).

### 2.5. Quantitative Reverse Transcription PCR

Real-time RT-qPCR assays were performed using 43 resected brain tumor sections. All samples were tested in triplicate. To quantify the results obtained by RT-PCR for *TGFβ1*, *TGFβ2*, *TGFβ3* and *ACTB*, a standard curve method was used, described previously by Strzalka-Mrozik et al. [9]. For the RT-qPCR reaction used Sensi-Fast™ reagent kit (Bioline, London, UK) and a set of primers (Forward and Reverse) with sequences complementary to the genes under study (Table 2). The gene encoding β-actin was used as a positive control of amplification The thermal conditions for the one-step RT-qPCR were as follows: reverse transcription at 45 °C, polymerase activation at 95 °C for 2 min, 40 duplicate cycles consisting of denaturation at 95 °C for 5 s and annealing at 60 °C for 10 s, then final elongation at 72 °C for 5 s. The Opticon™ DNA Engine Sequence Detector (MJ Research Inc., Watertown, MA, USA) used in this study plotted a standard curve based on fluorescence measurements for a known cDNA copy number—the quantitative template—and calculated the number of mRNA copies present in the reaction mixture. The final measure of the transcriptional activity of the genes tested was the mRNA copy number converted to 1 µg of total RNA. Each run was completed by melting curve analysis to confirm amplification specificity and the absence of primer dimers.

The specificity assessment of the RT-qPCR reaction was also confirmed by the 2% agarose gel electrophoresis technique.

### 2.6. Statistical Analysis

Data obtained from oligonucleotide microarrays were analyzed using the PL-Grid Infrastructure (http://www.plgrid.pl/; accessed on 29 March 2021). The GeneSpring 13.0 platform (Agilent Technologies UK Limited, South Queensferry, UK) and the Search Tool for the Retrieval of Interacting Genes/Proteins (STRING) database were used. An Excel spreadsheet (Microsoft Office Professional Plus 2016) was used to analyze the *TGFβ1*, *TGFβ2*, and *TGFβ3* mRNA copy number results obtained by RT-qPCR. Then, a database was created using Statistica software (StatSoft, Tulsa, OK, USA, version 13.1), and statistical analysis was performed. The normality of the distribution of the studied data was evaluated using the Shapiro–Wilk test. The non-parametric Mann–Whitney U test was used to determine the differences between the analyzed groups. The most significant descriptive statistics were determined for each analysis: median, upper, and lower quantiles, and minimum and maximum values. The significance level of *p* < 0.05 was assumed in statistical analyses. The presence of correlation between the studied variables was assessed on the basis of Spearman’s rank correlation coefficient (*r_s_*).

## 3. Results

### 3.1. Gene Expression Profile of TGFβ Isoforms Based on Oligonucleotide Microarray Analysis

In the first stage of the research, the expression profile of the three *TGFβ* isoforms was evaluated based on the results of the HG-U133A 2.0 oligonucleotide microarray. The microarray results were analyzed in the GeneSpring software, where the data were loaded as CEL files and automatically normalized in the software.

An analysis of the differentiation of the normalized values of fluorescent signals showed that the expression levels of *TGFβ1* and *TGFβ2* were statistically significantly changed in gliomas of different stages. In the case of the *TGFβ3* isoform, no statistically significant changes in expression levels were observed in the material studied.

Both *TGFβ1* and *TGFβ2* were overexpressed at stage G3/G4 relative to G2 in astrocytic brain tumors (Figure 1). The multiplicity of the change in the expression of the above-mentioned genes in the two test groups and the direction of the change were illustrated by the fold change (FC) parameter. A threefold higher expression of the *TGFβ1* isoform was observed in the G3/G4 stage compared with G2. Furthermore, *TGFβ2* was twice as highly expressed in the G3/G4 stage with respect to G2.

### 3.2. Gene Expression Profile of TGFβ Isoforms Based on RT-qPCR Analysis 

The results obtained with the oligonucleotide microarray method were validated using real-time RT-qPCR. The expression of the studied genes was presented as the number of mRNA copies converted to 1 μg of the total RNA.

Three *TGFβ* isoforms and *ACTB* were detected in all 43 glioma samples. The copy number of *TGFβ1* mRNA/μgRNA was found in group G2 and G3/G4 (Me = 6890.30 copies/μgRNA and Me = 14,736.70 copies/μgRNA, respectively); *TGFβ2* in group G2 and G3/G4 (Me = 3890.50 copies/μgRNA and Me = 9438.80 copies/μgRNA, respectively), *TGFβ3* in group G2 and G3/G4 (Me = 4772.10 copies/μgRNA and Me = 11,658.70 copies/μgRNA, respectively). The average numbers of *ACTB* mRNA were 152,288.40 and 286,668.70 copies of mRNA *ACTB*/μgRNA in groups G2 and G3/G4, respectively.

In the next part of the study, changes in the expression levels of *TGFβ1*, *TGFβ2*, *TGFβ3*, and *ACTB* were evaluated in gliomas with different degrees of malignancy. A statistically significant difference in *TGFβ2* expression was found between grades G3/G4 and G2 (Mann–Whitney U test; *p* = 0.039403). By contrast, *TGFβ1* and *TGFβ3* isoforms showed no statistically significant differences in their expression levels according to tumor malignancy grade (Mann–Whitney U test; *p* = 0.186948 and *p* = 0.163186, respectively) (Figure 2).

For the *ACTB* gene, no statistically significant differences were found in the expression levels between the analyzed groups (Mann–Whitney U test; *p* = 0.517009); therefore, this gene could be used as a reference gene. 

As a further step, a correlation analysis of the number of mRNA copies of individual *TGFβ* isoforms was performed, taking into account the division by grade of brain tumor malignancy. Using Spearman’s rank correlation coefficient, we found a positive, moderate correlation between the mRNA/1 μg RNA copy number of *TGFβ1* and that of *TGFβ2*, and between the mRNA/1 μg RNA copy number of *TGFβ2* and that of *TGFβ3* at malignancy grade G3/G4. Statistically significant correlations (*p* < 0.05) between individual parameters in the analyzed groups were as follows: *TGFβ1* versus *TGFβ2* of test group G3/G4 (*r_s_* = 0.539130; *p* = 0.006557) and *TGFβ2* versus *TGFβ3* of test group G3/G4 (*r_s_* = 0.457265; *p* = 0.016484) (Figure 3).

### 3.3. Assessment of Gene Expression Profiles and Their Relationships with TGFβ Isoforms

In the final step of this study, differences in the expression profiles of all genes on the HG-U133A 2.0 oligonucleotide microarray depending on the degree of cancer malignancy were analyzed, by generating a heatmap (Figure 4). The relationships of genes with *TGFβ* isoforms were also checked.

A different color pattern was observed on each of the heat maps, and thus it was confirmed that in both study groups (G2 versus G3/G4) the gene expression profiles differed. To select the differential genes, we performed a t-test. After the analysis, a Volcano plot showing the differential genes of the studied groups was generated.

For differentiating genes, those for which the FC parameter was greater than 2 were selected, with a statistical significance level of *p* < 0.05. Among the 22,283 probes present on the HG-U133A 2.0 oligonucleotide microarray, 1,402 probes were selected to distinguish between G3/G4 and G2 tumor malignancy (Table 3).

Afterward, genes showing altered expression in tumors of different stages were further characterized. 

Determination of the multiplicity of gene expression change was enabled by the FC parameter, indicating the log2 of the difference in fluorescence signals between G3/G4 versus G2 and the direction of the observed change.

After the differentiation criterion was narrowed down to FC > 10.0, from 1,402 probes only 16 characterized by the greatest variation in expression level and their corresponding genes and direction of expression change were selected (Table 4).

### 3.4. Analysis in the STRING Database

The relationships between previously selected genes in the STRING database were investigated. The analysis was aimed at verifying likely protein–protein interactions. Proteins encoded by the analyzed genes form a closely related network of probable protein–protein interactions consisting of 49 edges and 22 nodes (*p* < 0.001, medium confidence = 0.400). Edges represent the interactions between proteins, and the edge weight indicates the likelihood of interactions between them (Figure 5).

Subsequent analysis in the Kyoto Encyclopedia of Genes and Genomes (KEGG) and STRING databases revealed that the selected genes may be significantly involved in 236 biological processes and 27 signaling pathways, a significant number of which were related to tumorigenic processes (Table 5). 

## 4. Discussion

Brain tumors are a heterogeneous group of tumors because of the morphology of the cells from which they arise. In 2007, the WHO divided them into four grades of histological malignancy [27]. Astrocytomas are the most malignant primary tumors of the brain, among which the grade IV tumor, known as glioblastoma multiforme, is the most severe [28]. Advances in molecular biology techniques have allowed the characterization of the genome and the identification of epigenetic changes; in 2016, these advances resulted in the introduction of a new classification of the cancers in question based not only on histological criteria but mainly on gene analysis, which better grouped the cancers [5]. The aforementioned classification was updated in 2021 [6]. Given the differences between low- and high-grade gliomas in terms of histology, presenting symptoms, and gene expression profiles, molecular studies are undoubtedly important to aid in the clinical diagnosis and treatment of these tumors [29]. In particular, molecular studies on primary GBM have already identified 1473 genes, of which at least 43 showed significantly different levels of expression, depending on the survival time of patients. Moreover, a correlation was also found between tumor genotype and a patient’s length of survival [30]. 

The multifunctional cytokine TGFβ is involved in some key functions in the body. Among others, it is responsible for cell growth, differentiation, and migration and participates in repair and apoptosis [14]. The tumor process, including the development of gliomas, shows a pleiotropic character. At the beginning of the disease, it contributes to tumor suppression by controlling proliferation and inducing apoptosis. In advanced stages, it promotes metastasis and tumor progression [14,16]. Moreover, TGFβ inhibits the immune response against tumor cells. This effect avoids the death of tumor-transformed cells due to the activity of immune system elements, including T cells and NK cells [14,31].

The main aim of our study was to evaluate differences in the expression levels of transcripts associated with TGFβs in astrocytic brain tumors at different grades. We also quantified the mRNA of three *TGFβ* isoforms, which may prove helpful in diagnosis and in differentiating the grades of tumor lesions. Analysis of Affymetrix HG-U133A 2.0 oligonucleotide microarrays revealed 1402 differential genes between the G3/G4 group of highly advanced gliomas and the G2 group. A total of 16 genes were selected for further analysis, given FC > 10.0. The identified transcripts were characterized by overexpression in high-grade gliomas. In the next step, the analysis in the STRING and KEGG databases revealed the presence of significant probable protein–protein interactions consisting of 49 edges and 22 nodes (*p* < 0.001, medium confidence = 0.400) and that these proteins are involved in numerous biological processes and signaling pathways related to cancer progression. Gliomas of grade G2 can transform into more malignant tumors—G3 and G4. Van den Boom et al. [32], using oligonucleotide microarrays with 7129 probes, compared gene expression profiles in primary G2 gliomas and recurrent G3 and G4 gliomas. In their study, they used eight samples from patients with this type of cancer. The analyses carried out allowed the selection of 66 differentiating genes (*p* < 0.01; FC > 2), including the *COL4A2* gene (i.e., collagen chain type IV alpha 2). Using RT-qPCR, they quantified the expression of this gene in 43 samples of gliomas, and it was found to be positively correlated with the degree of malignancy. Furthermore, based on immunohistochemical analysis, the researchers found that higher expression of this gene in high-grade gliomas was associated with vascular proliferation in the tumors. Our study also showed significantly higher *COL4A2* gene expression in G3/G4 versus G2, which is consistent with the results obtained by Van den Boom et al. [32].

EGFR is a member of the tyrosine kinase superfamily receptor and is a potent oncogene. Alterations in *EGFR* expression are also observed in astrocytomas and may affect gliogenesis by promoting proliferation and modifying angiogenesis and invasion [33]. Waha et al. [34] assessed the amplification of the gene encoding *EGFR* by PCR in 97 glioma samples, including 26 at grade G2, 17 at grade G3, and 54 at grade G4. They observed no amplification in any of the samples at G2, whereas 6% of the G3 samples and 33% of the G4 samples showed *EGFR* amplification. Our study provided consistent results, as oligonucleotide microarray analysis showed increased expression of this gene in G3/G4 versus G2.

*CHI3L1* is involved in cell proliferation and differentiation and is possibly associated with tumor transformation. The protein product of this gene in patient plasma may be a useful prognostic marker in GBM [35]. Urbanavičiūtė et al. [36] used qRT-PCR to assess *CHI3LI1* expression in 20 tissue samples obtained from G2 astrocytoma patients and 24 GBM patient samples (G4), and in healthy human brain tissue. They observed increased expression in the G4 samples but decreased expression in the G2 samples relative to healthy brain tissue. This is partly consistent with our observations, as microarray analysis showed increased *CHI3L1* expression in G3/G4 versus G2 samples.

Protocadherins are members of the adhesion molecules responsible for cell–cell interactions. Waha et al. [37] identified a CpG island in the first exon of the *PCDHGA11* gene by microarray methylation analysis of astrocytomas that showed hypermethylation relative to normal tissue. Their results show that 82% of grade II astrocytomas, 80% of grade III astrocytomas and 74% of WHO grade IV gliomas showed non- or low expression of *PCDHGA11* compared to the average of five normal brain tissue samples. Similarly, the five glioma cell lines analyzed showed no or low expression of the aforementioned gene. However, interestingly, they also observed that *PCDHGA11* expression was significantly upregulated in several G4 samples, including and especially in recurrent glioma. They also confirmed that methylation of the mentioned fragment is associated with decreased gene transcription in stages G2 and G3, but such a correlation was not observed in stage IV glioma samples, where some showed high *PCDHGA11* transcription compared to normal tissue. This indicates that methylation of the study region may not be sufficient to silence transcription. This is partly consistent with our results, as we showed that *PCDHGA11* gene expression is higher in the G3/G4 versus G2 group.

In the available sources, we found no data on *COL6A2* expression in brain tumor tissues collected from patients at different stages of the disease. Thus, our study confirms the need for precise gene expression analysis in astrocytic brain tumors, as we demonstrated a statistically significant increase in *COL6A2* gene expression in G3/G4 relative to G2. Zhang et al. [38] attempted to identify potential genes related to breast cancer brain metastasis in breast cancer patients using bioinformatic tools and showed that several genes including *COL6A2* are negatively correlated with survival, which in a way confirms our results indicating overexpression of this gene in tumors of higher malignancy grade. 

*SEC61G* is overexpressed in gastric and breast cancers. Liu et al. [39] using the Cancer Genome Atlas and the Chinese Glioma Genome Atlas datasets evaluated the correlation between the expression of this gene and the survival prognosis of patients with glioblastoma multiforme. They observed that overexpressing *SEC61G* was associated with short patient survival. Our data from oligonucleotide microarray analysis indicate that *SEC61G* expression is higher in malignant brain tumors, which is also indirectly associated with survival. 

In our study, we observed increased expression of *EMP3* and *TIMP1* in patients with higher-grade astrocytic brain tumors. These genes were also evaluated in several other studies. Wang et al. [40] evaluated the potential prognostic significance of age-related genes in patients with lower-grade glioma. Using bioinformatics tools and available databases, they demonstrated that *EMP3* and *TIMP1* are overexpressed in tumor tissue than in control normal brain tissue and that increased expression of these genes is associated with poorer prognosis for patients. Similarly, Zhang et al. [41] using bioinformatics tools, databases and clinical data assessed *EMP3* expression in samples from glioma patients. They also demonstrated that increased expression is observed in patients with a higher degree of disease, which is consistent with our results. Zhang et al. also demonstrated an association between *EMP3* expression and prognosis of survival. 

POSTN is a protein mainly associated with the extracellular matrix. Its expression is mainly observed in cancer cells including glioblastoma multiforme at the last stage of tumorigenesis by promoting mesenchymal-epithelial transition. It is also expressed in the stroma of normal stem cells, contributing to increased invasiveness [42]. The study by Huizer et al. [43] examined *POSTN* expression levels in glioma cells at different stages of patient samples. Using molecular biology techniques, they found a significant difference in *POSTN* expression depending on the stage of the disease. In Stage G2, the expression level was significantly lower, and its increase correlated with the stage of the disease. Our observations also indicate increased *POSTN* expression in G3/G4 stage patients compared to lower stages.

Overexpression of *IGFBP2* gene is observed in many neoplastic diseases including glioblastoma multiforme. Due to its characteristic structure, it can bind with integrin-promoting processes associated with migration and invasion of tumor cells. Additionally, it also plays an important role during the developmental stage of the brain, mainly in fetal tissues, and its expression decreases with age. A significant increase after the developmental stage is observed mainly in pathological states of the brain, associated with trauma or hypoxia, as well as in tumor occurrence [44]. Fuller et al. [45] evaluated with oligonucleotide microarrays the expression level of *IGFBP2* in 24 tissue samples of gliomas at different stages of disease. An increase in *IGFBP2* expression is positively correlated with the stage of the disease, with the highest expression in stage G4. The study was further extended by analysis at the protein level, which confirmed the observed change in expression. In our study, we also showed a significant statistical increase in *IGFBP2* gene expression between stages G3/G4 and G2. This may suggest that *IGFBP2* is involved in processes related mainly to the migration and metastasis of tumor cells in the most advanced stage of gliomas.

EIF5A is a protein that undergoes a specific post-translational modification called hypusin and it is highly conserved. This protein is mainly involved in cell proliferation and survival and is also involved in the regulation of programmed cell death, both in normal and neoplastic cells, including glioblastoma multiforme. In a study by Preukschas et al. [46] using sections from 173 brain tumors at various stages of development, *EIF5A* expression levels were assessed by immunohistochemistry. Researchers observed an increase in *EIF5A* expression independent of the disease stage. In our study using oligonucleotide microarrays, however, we showed that *EIF5A* expression varies and depends on the stage of the disease. Stage G3/G4 gliomas show higher expression of *EIF5A* than lower stage G2 gliomas.

SERPINA3 is mainly involved in processes related to cell survival and proliferation, and plays a key role in various types of cancer. Increased expression is observed not only in brain tumors, but also in colorectal cancer, endometrial cancer, melanoma or breast cancer [47]. Our study revealed that the expression level of *SERPINA3* was higher in the highest grade samples compared to the lower grade of malignancy. Luo et al. [47] conducted a similar study; however, they only assessed *SERPINA3* expression levels by immunohistochemistry and RT-qPCR on samples from 180 patients with varying degrees of glioma. The results at the mRNA and protein levels showed that *SERPINA3* expression was correlated with the stage of disease.

Annexin 1 plays an important role in proliferation and apoptosis. It also has anti-inflammatory properties. Reduced expression levels of *ANXA1* have so far been described in several types of tumors including breast and head and neck cancers. In our study on brain tumor samples from patients, *ANXA1* expression levels were assessed using molecular biology techniques and oligonucleotide microarrays and showed an increase in expression of this gene in brain tumors with the highest malignancy, namely glioblastoma multiforme [48]. A study conducted by Ruano et al. [49] on a group of 20 also observed a significant increase in *ANXA1* expression, which was assessed by RT-qPCR and oligonucleotide microarrays. In our study, we also observed an increase in the expression level of *LTF* in patients with stage G3/G4 glioma compared to G2. LTF is classified as a transcriptional factor and one of its key roles is to participate in cell proliferation and growth. In a study performed by Tyburczy et al. [48] on samples from 10 patients with subependymal giant cell astrocytomas, mTOR-regulated proteins were assessed. The researchers obtained similar results using oligonucleotide microarrays, indicating that *LTF* expression is increased in brain tumors.

PCDHGA3 is a protein belonging to protocadherin. It is mainly involved in the cellular aging process. Until now the expression level of this gene has only been reported in non-cancer-related studies [50]. In our study, we focus on gliomas, which are brain tumors. In microarray analysis, we observed an increase in the expression of the gene encoding PCDHGA3 at the G3/G4 versus G2 stage. This suggests that *PCDHGA3* is also up-regulated in neoplastic cells and therefore further studies on the possible involvement of *PCHDGA3* in the neovascularization process are needed.

Among the TGFβ family are six isoforms, of which only three are present in humans [7]. Based on the results of the oligonucleotide microarray HG-U133A 2.0 analysis, an almost threefold increase in *TGFβ1* and *TGFβ2* isoform expression was found in the G3/G4 versus G2 group. By contrast, changes in *TGFβ3* gene expression were not statistically significant. The analyses performed in this study concluded that the expression of *TGFβ1* and *TGFβ2* increased with increasing tumor grade and, thus, malignancy.

As previously mentioned, TGFβ can influence immune cell response. Double immunohistochemical staining was used by Zhang et al. [24] to investigate the correlation between *TGFβ1* expression levels and the number of regulatory T (Treg) cells. The present study was performed on 135 samples of gliomas at four grades (WHO grades I–IV) and on 15 healthy brain tissues. Increased *TGFβ1* expression was observed in the majority of glioma samples and correlated with a high grade, which is consistent with the results obtained in our work. Furthermore, this expression was positively associated with the number of Treg cells (*p* < 0.01) in high-grade gliomas. Tumor-produced TGFβ1 isoform may be associated with Treg cell infiltration in glioma tissues, and high levels are indicative of a poor prognosis. Treg lymphocytes prevent the proliferation of T cells (CD8^+^), thereby leading to the inhibition of the release of cytotoxic cytokines and chemokines, resulting in the inhibition of the anti-tumor immune response [51]. TGFβ also affects the induction of epithelial–mesenchymal transition [17]. Yang et al. [25] examined the expression of *TGFβ1* and E-cadherin in gliomas and healthy brain tissues. They searched for correlations between their expression and pathological features and their clinical significance. RT-PCR and Western blotting were used to conclude that *TGFβ1* expression (*p* < 0.01) was higher in gliomas compared with brain tissues without pathological changes. Furthermore, low-differentiated tumors had lower expression than well-differentiated gliomas (*p* < 0.01). The lower the degree of cell differentiation and the more malignant tumor forms, the higher *TGFβ1* expression levels were observed. A similar correlation in the expression level of the *TGFβ1* isoform with respect to the grade of malignancy of the brain tumor was shown in our study, but the correlation was not statistically significant.

Numerous changes have been observed in tumor cells, one of which may be the inability to undergo apoptosis, which is associated with treatment resistance, among other things. Mabrouk et al. [52] studied the effect of apoptosis signals including TGFβ1 in the cytoplasm with survival rate in 30 grade II, III and IV astrocytic tumors. Using ELISA enzyme immunoassay, they showed that cytosolic TGFβ1 levels did not vary statistically significantly with the malignancy stage, but higher TGFβ1 levels were associated with longer survival. The results obtained by Mabrouk et al. are not consistent with ours or those of the previously mentioned authors, suggesting an increase in levels of this isoform in advanced brain tumors. The discrepancies may be due to the high heterogeneity of brain tumors between patients, the size of the study group, or the fact that we used other molecular methods that are more specific than protein concentration assessment. Undoubtedly, the question of the correlation between TGFβ1 isoform levels and the grade of astrocytic brain tumors requires further comprehensive studies. 

However, the *TGFβ2* isoform deserves the most attention, as the results of real-time RT-qPCR demonstrated that this cytokine showed a statistically significant difference in expression levels between the groups studied (G3/G4 versus G2). The results obtained are in line with those of Kjellman et al. [26], who, in a study on 23 glioma samples, showed that *TGFβ2* mRNA expression was significantly increased (*p* < 0.005) and that its increase correlated with increasing malignancy. They selected three glioblastomas (G2), eight anaplastic gliomas (G3), twelve GBM (G4), and healthy brain tissue for the study. Analysis was performed using RT-qPCR. In the case of *TGFβ1* (*p* < 0.001) and *TGFβ3* (*p* < 0.019) isoforms, the researchers found a significant increase in expression, but not as strong as for *TGFβ2* isoforms. These relationships were not noted in our study, as no statistically significant differences were found in their expression levels depending on the degree of tumor malignancy (*p* > 0.05). According to the aforementioned researchers, both TGFβ1 and TGFβ2 may be important in the early grades of tumor development, although only the TGFβ2 isoform is particularly important in more advanced grades, which can also be related to our research.

Cytokines present in the tumor microenvironment, including TGFβ2, may influence tumor cell invasion. Zhang et al. [20] investigated the relationship between *TGFβ2* expression levels and autophagy in gliomas. The process of autophagy is an important factor in tumor cell aggressiveness, but the effect of this cytokine on the process in question is poorly understood. The researchers used cell lines and samples from patients with GBM. Analyses were performed using Western blotting, qPCR, and immunofluorescence, among other techniques. Student’s *t*-test used by the researchers found a difference in *TGFβ2* expression between GBM samples and healthy brain tissues (*p* < 0.0490). Zhang et al. [20] further found that high *TGFβ2* expression simultaneously correlated with poor prognosis and short survival times, whereas it was unrelated to patient age and gender. In our study, *TGFβ2* mRNA expression levels were also positively correlated with the degree of malignancy, but the results were not compared with healthy tissue.

*TGFβ2* plays a key role in many biological processes, some of which are associated with tumorigenesis. This was also confirmed by our own research. Among other things, this cytokine participates in the epithelial-mesenchymal transition process, which underlies tumor development and progression. Moreover, it is connected with the response to growth factors, tissue development or positive regulation of cell population proliferation and migration, which may also be connected with tumor induction. *TGFβ2* also affects the negative regulation of macrophage cytokine production, which may be related to a reduced immune response within the tumor. In view of this, the key role of *TGFβ2* in tumor progression and development seems to be undeniable. The association of this isoform with processes involved in tumor development, progression and invasion are also consistent with our results showing higher expression of this cytokine in stage G3/G4 relative to G2.

TGFβ is a cytokine whose involvement in the development of many cancers has already been documented, including melanoma, lung cancer, pancreatic cancer, breast cancer, and colorectal cancer [19]. Javle et al. [21] investigated TGFβ1 plasma levels in patients with pancreatic ductal adenocarcinoma and showed that an increase in its levels was accompanied by a short survival time. Chen et al. [22] revealed high *TGFβ1* mRNA levels in breast cancer tissues compared with healthy tissue. By contrast, Bellone et al. [53] presented that colon cancer progression was accompanied by an increase in the mRNA expression of both *TGFβ1* and *TGFβ2*. Furthermore, the increase in plasma levels of these cytokines was greater in cancer patients than in healthy individuals. However, further research on the specificity of the activity of this cytokine in cancer is needed. Notably, with the ever-increasing importance of the oncogenic role of TGFβ as a tumor promoter, interest in targeting this cytokine in therapy is growing among researchers [16]. In particular, the use of drugs that act directly on TGFβ signaling appears to be a promising therapeutic target.

To summarize, our results are mostly consistent with those obtained by other researchers. Our study has the advantage of performing assays in astrocytic brain tumor tissues using several molecular techniques simultaneously, which allowed us to distinguish changes in the expression of transforming growth factor β isoforms and other genes depending on the tumor grade. Moreover, the data obtained from the bioinformatics analysis showed that the genes studied actually play a key role in the signaling pathways and biological processes involved in carcinogenesis, confirming that their overexpression at the G3/G4 stage may indeed be associated with malignancy and tumor invasion.

The expression levels of the three *TGFβ* isoforms in astrocytic brain tumors, as in other tumors, may be a useful diagnostic and prognostic marker in the future. To this end, additional research on a larger group is needed to accurately characterize methodologies for markers in brain tumors, but this study may provide a valuable basis.

The correlation found between the tumor grade and the expression level of these growth factors may be helpful in predicting the course of the disease. In particular, biomarkers of favorable prognosis in GBM are still being sought in large-scale studies. In the future, analysis of the expression profile of genes involved in TGFβ signaling may also prove useful in developing new treatment strategies.

## 5. Conclusions

Changes in the expression pattern of transcripts associated with *TGFβ* isoforms in astrocytomas classified by malignancy confirm their important role in tumor development and progression. A quantitative assessment of *TGFβ2* mRNA in patients suffering from brain tumors may facilitate the identification of the stage of the lesion and be a potential method to complement the diagnostic process. However, further research is required to develop specific protocols that could be widely used in the diagnosis of brain tumors.

## Figures and Tables

**Figure 1 cancers-14-01876-f001:**
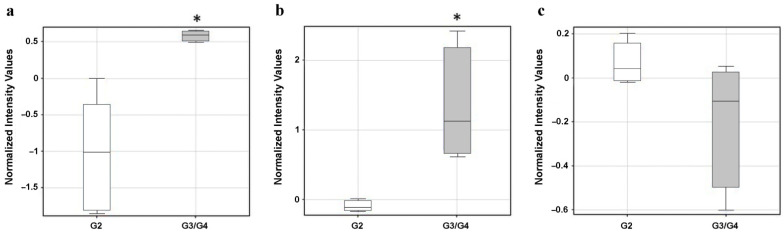
*TGFβ1* (**a**); *TGFβ2* (**b**) and *TGFβ3* (**c**) expression profiles in relation to tumor grade obtained by oligonucleotide microarray. Results are presented as median with lower and upper quartiles and minimum and maximum values; G2, G3/G4—WHO grade of malignancy; * —statistical statistical significance (*p* < 0.05).

**Figure 2 cancers-14-01876-f002:**
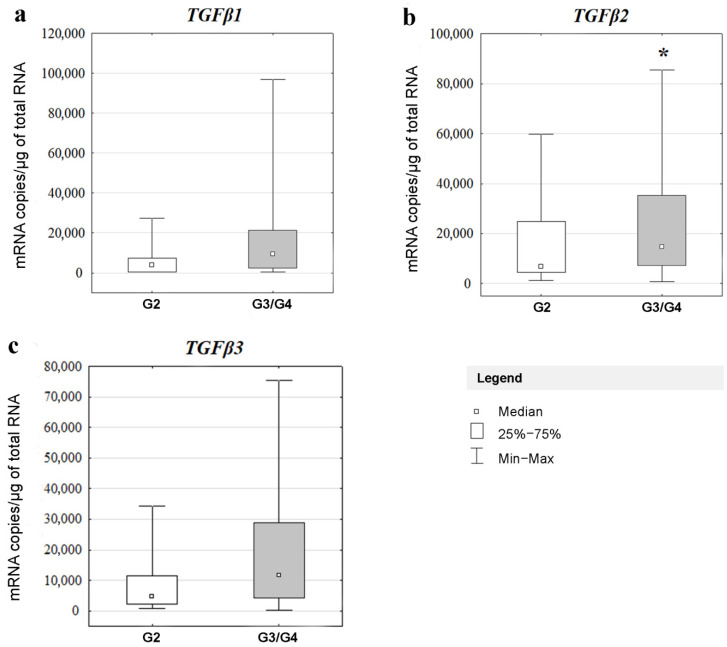
Expression of *TGFβ* isoforms in gliomas with different grades of malignancy obtained by RT-qPCR. Results are presented as median with lower and upper quartiles and minimum and maximum values; G2, G3/G4—WHO grade of malignancy; *—statistical significance (*p* < 0.05); (**a**) *TGFβ1*; (**b**) *TGFβ2* and (**c**) *TGFβ3*.

**Figure 3 cancers-14-01876-f003:**
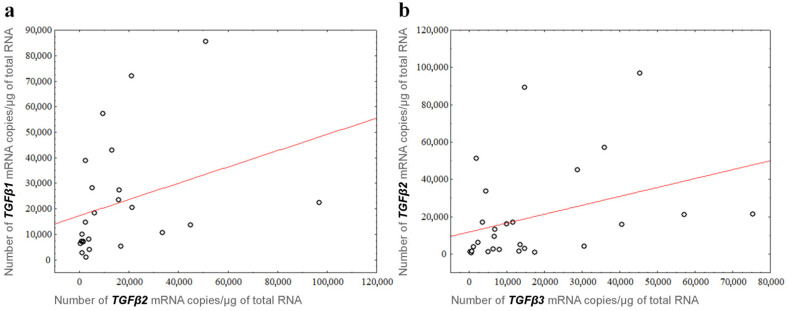
*TGFβ2* expression is positively correlated with (**a**) *TGFβ1* (*r_s_* = 0.539130) and (**b**) *TGFβ3* (*r_s_* = 0.457265) in gliomas with G3/G4 grades of malignancy.

**Figure 4 cancers-14-01876-f004:**
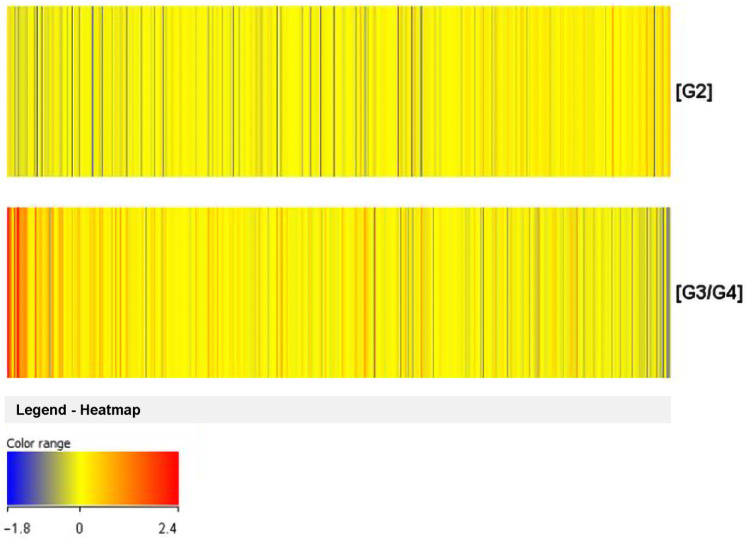
Graphical illustration of differences in expression profiles genes based on tumor malignancy. The color variation between transcriptome groups indicates the presence of differences in gene expression profiles depending on tumor malignancy. Red—higher signal; high gene expression; Blue—lower signal; low gene expression; G2, G3/G4—grade of tumor malignancy.

**Figure 5 cancers-14-01876-f005:**
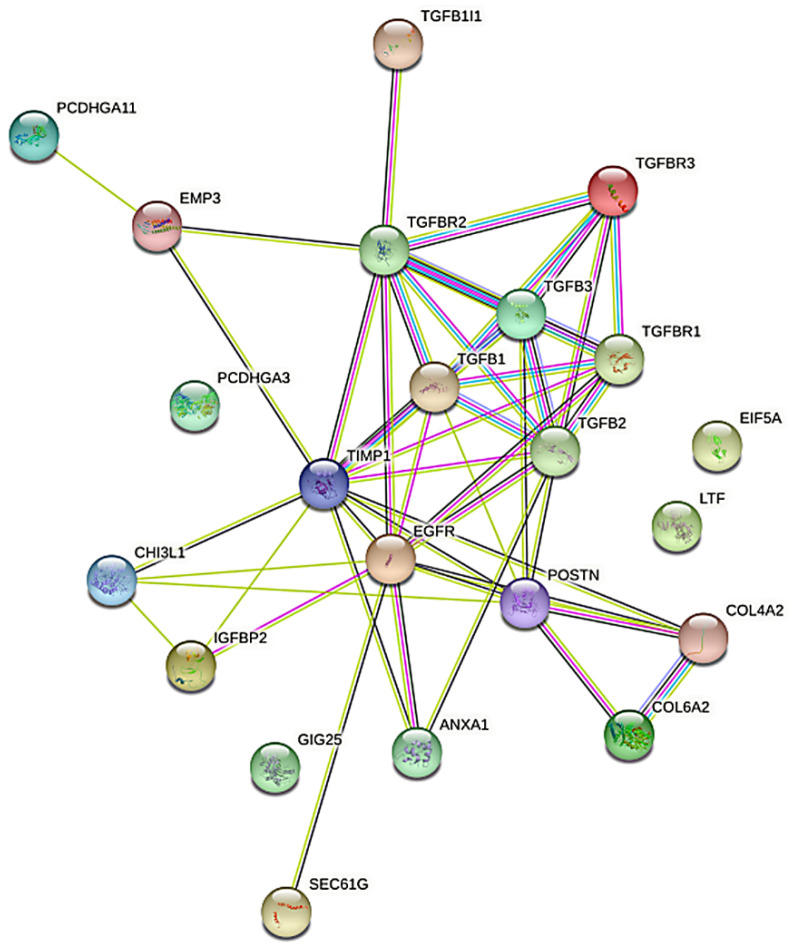
Protein–protein interaction network generated in the STRING database. STRING database—Search Tool for the Retrieval of Interacting Genes/Proteins.

**Table 1 cancers-14-01876-t001:** Selected clinical features of the studied group of patients.

Gender	Age (Yrs)	WHO Grade of Malignancy	Number of Samples
Female (*n* = 21)	56 ± 13	G2	4
		G3	2
		G4	15
Male (*n* = 22)	52 ± 14	G2	8
		G3	3
		G4	11

G2—II grade; G3—III grade; G4—IV grade; WHO—World Health Organization; values of clinical parameters are expressed as means ± standard deviation.

**Table 2 cancers-14-01876-t002:** Sequences of primers used in the RT-qPCR reaction.

Gene	Oligonucleotide Sequence	Amplimer Length (bp)	Tm (°C)
*TGFβ1*	Forward: 5′TGAACCGGCCTTTCCTGCTTCTCATG3′ Reverse: 5′GCGGAAGTCAATGTACAGCTGCCGC3′	152	87.4
*TGFβ2*	Forward: 5′TACTACGCCAAGGAGGTTTACAAA3′ Reverse: 5′TTGTTCAGGCACTCTGGCTTT3′	201	88.2
*TGFβ3*	Forward: 5′CTGGATTGTGGTTCCATGCA3′ Reverse: 5′TCCCCGAATGCCTCACAT3′	121	82.7
*ACTB*	Forward: 5′TCACCCACACTGTGCCCATCTACGA3′ Reverse: 5′CAGCGGAACCGCTCATTGCCAATGG3′	295	88.2

Tm—melting temperature; bp—base pairs.

**Table 3 cancers-14-01876-t003:** Number of probes differentiating the test groups according to statistical assumptions.

		Total *p*	*p* < 0.05	*p* < 0.02	*p* < 0.01	*p* < 0.005	*p* < 0.001
G3/4 vs.G2							
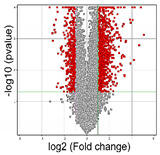	Number of probes	22283	6378	3991	2760	1856	670
	FC > 1.1	14436	6304	3970	2754	1855	670
	FC > 1.5	4211	3341	2580	1979	1450	573
	FC > 2.0	1569	1402 *	1186	1008	819	376
	FC > 3.0	420	390	356	320	277	143

FC—fold change; *—statistical significance (*p* < 0.05); FC > 2.0; G2, G3/G4—grade of tumor malignancy.

**Table 4 cancers-14-01876-t004:** Characteristics of genes showing altered expression in gliomas of different grades.

Probe	Gene Symbol	Gene Name	FC G3/G4 vs. G2	Expression Change G3/G4 vs. G2
209396_s_at	*CHI3L1*	Chitinase-3-like protein 1	33.22	↑
202718_at	*IGFBP2*	Insulin Like Growth Factor Binding Protein 2	27.72	↑
209395_at	*CHI3L1*	Chitinase-3-like protein 1	24.87	↑
211876_x_at	*PCDHGA11*	Protocadherin Gamma Subfamily A, 11	18.84	↑
210809_s_at	*POSTN*	Periostin	18.27	↑
209156_s_at	*COL6A2*	Collagen Type VI Alpha 2 Chain	16.79	↑
201012_at	*ANXA1*	Annexin A1	16.41	↑
201666_at	*TIMP1*	TIMP Metallopeptidase Inhibitor 1	14.81	↑
201983_s_at	*EGFR*	Epidermal Growth Factor Receptor	14.75	↑
203729_at	*EMP3*	Epithelial Membrane Protein 3	14.28	↑
211966_at	*COL4A2*	Collagen Type IV Alpha 2 Chain	11.10	↑
203484_at	*SEC61G*	SEC61 Translocon Subunit Gamma	10.93	↑
202018_s_at	*LTF*	Lactotransferrin	10.26	↑
201123_s_at	*EIF5A*	Eukaryotic translation initiation factor 5A-1	10.14	↑
216352_x_at	*PCDHGA3*	Protocadherin Gamma Subfamily A, 3	10.12	↑
202376_at	*SERPINA3*	Serpin Family A Member 3	10.10	↑

G2, G3/G4—WHO grade of malignancy; FC—fold change; ↑—gene overexpression; statistical significance (*p* < 0.05).

**Table 5 cancers-14-01876-t005:** Selected signaling pathways and biological processes in which specific differential genes play an important role.

**Signal Pathways**	**Gene**	***p* Value ***
TGF-beta signaling pathway	*TGFB1*, *TGFB3*, *TGFBR2*, *TGFBR1*	<0.001
FoxO signaling pathway	*TGFB1*, *TGFB3*, *EGFR*, *TGFBR2*, *TGFBR1*	<0.001
Relaxin signaling pathway	*TGFB1*, *EGFR*, *TGFBR2*, *COL4A2*, *TGFBR1*	<0.001
AGE-RAGE signaling pathway in diabetic complications	*TGFB1*, *TGFB3*, *TGFBR2*, *COL4A2*, *TGFBR1*	<0.001
MAPK signaling pathway	*TGFB1*, *TGFB3*, *EGFR*, *TGFBR2*, *TGFBR1*	<0.001
Pathways in cancer	*TGFB1*, *TGFB3*, *EGFR*, *TGFBR2*, *COL4A2*, *TGFBR1*	<0.001
Hippo signaling pathway	*TGFB1*, *TGFB3*, *TGFBR2*, *TGFBR1*	<0.001
**Biological Process**	**Gene**	***p* Value ***
**Cellular response to transforming growth factor beta stimulus**	*TGFBR3*, *TGFB1*, *TGFB3*, *TGFBR2*, *COL4A2*, ***TGFB2***, *TGFBR1*, *POSTN*, *TGFB1I1*	<0.001
**Cellular response to growth factor stimulus**	*TGFBR3*, *TGFB1*, *TGFB3*, *EGFR*, *TGFBR2*, *COL4A2*, ***TGFB2***, *TGFBR1*, *ANXA1*, *POSTN*, *TGFB1I1*	<0.001
**Pathway-restricted SMAD protein phosphorylation**	*TGFBR3*, *TGFB1*, *TGFBR2*, ***TGFB2***, *TGFBR1*	<0.001
**Transforming growth factor beta receptor signaling pathway**	*TGFBR3*, *TGFB1*, *TGFB3*, *TGFBR2*, ***TGFB2***, *TGFBR1*, *TGFB1I1*	<0.001
**Positive regulation of epithelial to mesenchymal transition**	*TGFB1*, *TGFB3*, *TGFBR2*, ***TGFB2***, *TGFBR1*, *TGFB1I1*	<0.001
**Positive regulation of cell population proliferation**	*TGFBR3*, *TIMP1*, *TGFB1*, *LTF*, *IGFBP2*, *TGFB3*, *EGFR*, *EIF5A*, *TGFBR2*, *TGFB2*, *TGFBR1*, *ANXA1*	<0.001
Negative regulation of transforming growth factor beta receptor signaling pathway	*TGFBR3*, *TGFB1*, *TGFB3*, *TGFBR2*, *TGFBR1*, *TGFB1I1*	<0.001
**Response to endogenous stimulus**	*TGFBR3*, *TIMP1*, *TGFB1*, *IGFBP2*, *TGFB3*, *EGFR*, *TGFBR2*, *COL4A2*, ***TGFB2***, *TGFBR1*, *ANXA1*, *POSTN*,	<0.001
**Regulation of cell population proliferation**	*TGFBR3*, *TIMP1*, *TGFB1*, *LTF*, *IGFBP2*, *TGFB3*, *EGFR*, *EIF5A*, *TGFBR2*, ***TGFB2***, *TGFBR1*, *ANXA1*, *TGFB1I1*	<0.001
**Tissue development**	*TGFBR3*, *TIMP1*, *TGFB1*, *CHI3L1*, *EGFR*, *COL6A2*, *TGFBR2*, *COL4A2*, ***TGFB2***, *TGFBR1*, *ANXA1*, *POSTN*, *TGFB1I1*	<0.001
**Positive regulation of transmembrane receptor protein serine/threonine kinase signaling pathway**	*TGFBR3*, *TGFB1*, *TGFB3*, ***TGFB2***, *TGFBR1*, *TGFB1I1*	<0.001
**Enzyme linked receptor protein signaling pathway**	*TGFBR3*, *TGFB1*, *TGFB3*, *EGFR*, *TGFBR2*, *COL4A2*, ***TGFB2***, *TGFBR1*, *TGFB1I1*	<0.001
**Negative regulation of macrophage cytokine production**	*TGFB1*, *TGFB3*, ***TGFB2***	<0.001
**Regulation of developmental process**	*TGFBR3*, *TIMP1*, *TGFB1*, *LTF*, *TGFB3*, *CHI3L1*, *EGFR*, *TGFBR2*, *COL4A2*, ***TGFB2***, *TGFBR1*, *ANXA1*, *POSTN*, *TGFB1I1*	<0.001
**Positive regulation of protein kinase activity**	*TGFB1*, *LTF*, *TGFB3*, *CHI3L1*, *EGFR*, *TGFBR2*, ***TGFB2***, *TGFBR1*	<0.001
**Positive regulation of pathway-restricted SMAD protein phosphorylation**	*TGFB1*, *TGFB3*, ***TGFB2***, *TGFBR1*	<0.001
**Negative regulation of signal transduction**	*TGFBR3*, *TGFB1*, *LTF*, *IGFBP2*, *TGFB3*, *EGFR*, *TGFBR2*, ***TGFB2***, *TGFBR1*, *TGFB1I1*	<0.001
**Negative regulation of cell differentiation**	*TGFB1*, *LTF*, *EGFR*, ***TGFB2***, *TGFBR1*, *ANXA1*, *POSTN*, *TGFB1I1*	<0.001
**Positive regulation of cell migration**	*TGFB1*, *EGFR*, *TGFBR2*, ***TGFB2***, *TGFBR1*, *ANXA1*, *POSTN*	<0.001
Positive regulation of SMAD protein signal transduction	*TGFB1*, *TGFB3*, *TGFBR1*	<0.001
**Epithelial to mesenchymal transition**	*TGFBR3*, *TGFB1*, ***TGFB2***, *TGFBR1*	<0.001
**Regulation of cell communication**	*TGFBR3*, *TIMP1*, *TGFB1*, *LTF*, *IGFBP2*, *TGFB3*, *CHI3L1*, *EGFR*, *TGFBR2*, ***TGFB2***, *TGFBR1*, *ANXA1*, *POSTN*, *TGFB1I1*	<0.001
**Regulation of cell migration**	*TIMP1*, *TGFB1*, *EGFR*, *TGFBR2*, ***TGFB2***, *TGFBR1*, *ANXA1*, *POSTN*	<0.001
**Regulation of signaling**	*TGFBR3*, *TIMP1*, *TGFB1*, *LTF*, *IGFBP2*, *TGFB3*, *CHI3L1*, *EGFR*, *TGFBR2*, ***TGFB2***, *TGFBR1*, *ANXA1*, *POSTN*, *TGFB1I1*	<0.001
**Positive regulation of cell differentiation**	*TGFB1*, *LTF*, *TGFB3*, *TGFBR2*, ***TGFB2***, *TGFBR1*, *ANXA1*, *TGFB1I1*	<0.001
**Cell adhesion**	*PCDHGA3*, *EGFR*, *COL6A2*, *TGFBR2*, ***TGFB2***, *ANXA1*, *POSTN*, *TGFB1I1*, *PCDHGA11*	<0.001
**Regulation of cell differentiation**	*TGFB1*, *LTF*, *TGFB3*, *EGFR*, *TGFBR2*, ***TGFB2***, *TGFBR1*, *ANXA1*, *POSTN*	0.0016
**Positive regulation of epithelial cell migration**	*TGFB1*, *TGFBR2*, ***TGFB2***, *ANXA1*	0.0022
**SMAD protein signal transduction**	*TGFB1*, *TGFB3*, ***TGFB2***	0.0036
**Establishment of localization in cell**	*TIMP1*, *TGFB1*, *LTF*, *TGFB3*, *PCDHGA3*, *CHI3L1*, *EIF5A*, ***TGFB2***, *GIG25*, *SEC61G*	0.0078
**Cell death**	*TGFB1*, *CHI3L1*, *EMP3*, *EIF5A*, *TGFBR2*, ***TGFB2***, *TGFBR1*	0.0091

STRING database—Search Tool for the Retrieval of Interacting Genes/Proteins; *—statistical significance (*p* < 0.05); the processes in which the *TGFβ2* isoform plays an important role are indicated in bold.

## Data Availability

The data presented in this study are available in this article.

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
