# Peer review of "Differences in the Expression Patterns of TGFβ Isoforms and Associated Genes in Astrocytic Brain Tumors"

_cancers, 2022, doi:10.3390/cancers14081876_

Round 1

Reviewer 1 Report

In the proposed manuscript, the authors present interesting results concerning the altered expression of TGFβ isoforms encoding genes, and their potential impact on increased tumor malignancy, i.e., G3/4 stage in comparison with G1. The main achievement of the study is a conclusion that quantitative evaluation of TGFβ2 mRNA can be adapted as a potential diagnostic marker useful for brain tumors malignancy grade assessment. However, further studies are still needed. Obtained results were supported by an additional database search to identify TGFβ related genes, which altered expression could increase the tumorigenic processes in the brain. Generally, the manuscript is well organized and detailed except that some sections need rewording and rephrasing to a better understanding of the concept. I have a few comments that need to be addressed by the author.

  1. Please include in the abstract the p values where needed. Providing such information would give potential readers to get brief information with no need to look it for in the manuscript body.
  2. Include in the Introduction section the broad characteristics of the new molecular classification of brain tumors based on genome characterization and epigenetic changes.
  3. Inconsistency in using genes symbols. Human genes should be written using capital letters and italics. Please follow this in the whole manuscript body, to make it clear, where the authors mention mRNA or protein levels of discussed molecules.
  4. The Results section would benefit from a graphical presentation of correlation analysis of TGFβ isoforms expression level and tumor malignancy grade.
  5. The font size in the figures is too small and sometimes it is difficult to read. I think that there is enough space to increase the font size in most of the figures. Compare with relatively large Tables.
  6. Figure 1 should be improved to increase its clarity. In the current form, it looks like it is cropped in the wrong way.
  7. In my opinion, the statistical differences (*) should be placed above the G3/G4 bars in the graphs. G2 serves here as a control in statistical analysis.
  8. I am not fully convinced about using the ACTB gene as endogenous control. For instance, the group G3/G4 displays an almost 2-fold higher copy number than in the case of the G2 group. I noticed this issue is discussed in the further part of the Results section, and as it is stated the ACTB gene copy number did not show significant differences between analyzed groups. Did the authors use the ACTB gene for data normalization?
  9. In Figure 3, the authors could include a legend describing the fold-change of differentially expressed genes.
  10. Please avoid citing scientific papers published in other languages than English (see citation 11).

Author Response

Comments:

In the proposed manuscript, the authors present interesting results concerning the altered expression of TGFβ isoforms encoding genes, and their potential impact on increased tumor malignancy, i.e., G3/4 stage in comparison with G1. The main achievement of the study is a conclusion that quantitative evaluation of TGFβ2 mRNA can be adapted as a potential diagnostic marker useful for brain tumors malignancy grade assessment. …”

Answer:

We would like to thank the Reviewer for notice of value of our research. We tried to take into account all the comments of the Reviewer improving the quality of our work.

Comments:

Moderate English changes required”

Answer:

Manuscript was revised extensively. American English Editor refined and corrected our manuscript to make it more clear, readable, and grammatically correct. We used Scribendi (online Web-based professional copy editing company). Editor code - EM2189.

Comments:

Does the introduction provide sufficient background and include all relevant references? Can be improved (x)”

Answer:

The introduction section has been updated as suggested by both Reviewers.

Comments:

Are the results clearly presented? Can be improved (x)”

Answer:

The results section was graphically modified. We hope that this part of manuscript in its current form is clearer.

Comments:

“Please include in the abstract the p values where needed. Providing such information would give potential readers to get brief information with no need to look it for in the manuscript body.”

Answer:

Included in the abstract of the p values as suggested by the Reviewer.

Comments:

“Include in the Introduction section the broad characteristics of the new molecular classification of brain tumors based on genome characterization and epigenetic changes.”

Answer:

Thankfully to Reviewer’s suggestion we added the appropriate information.

Comments:

“Inconsistency in using genes symbols. Human genes should be written using capital letters and italics. Please follow this in the whole manuscript body, to make it clear, where the authors mention mRNA or protein levels of discussed molecules.”

Answer:

Mentioned mistake has been corrected. We apologize for this situation.

Comments:

“The Results section would benefit from a graphical presentation of correlation analysis of TGFβ isoforms expression level and tumor malignancy grade.”

Answer:

Graphical presentation of the analysis of the correlation of the expression level of TGFβ isoforms and the degree of malignancy of the tumor has been included in the Results section.

Comments:

“The font size in the figures is too small and sometimes it is difficult to read. I think that there is enough space to increase the font size in most of the figures. Compare with relatively large Tables.”

Answer:

The font size in most figures has been increased whenever possible.

Comments:

“Figure 1 should be improved to increase its clarity. In the current form, it looks like it is cropped in the wrong way.”

Answer:

Figure 1 presented in the manuscript was generated automatically in the GeneSpring program. The graphic form has not been changed by the operator.

The second version of the Figure 1 can be as follows and perhaps it more clarity.

The Figure 1 in the results section have been completed and modified. We hope it will make it more pleasant.

Comments:

“In my opinion, the statistical differences (*) should be placed above the G3/G4 bars in the graphs. G2 serves here as a control in statistical analysis.”

Answer:

As suggested by the Reviewer, the statistical differences (*) are shown above the G3/G4 bars in the graphs.

Comments:

“I am not fully convinced about using the ACTB gene as endogenous control. For instance, the group G3/G4 displays an almost 2-fold higher copy number than in the case of the G2 group. I noticed this issue is discussed in the further part of the Results section, and as it is stated the ACTB gene copy number did not show significant differences between analyzed groups. Did the authors use the ACTB gene for data normalization?”

Answer:

We also were not fully convinced of the use of the ACTB gene as an endogenous control.Indeed, we notice that the ACTB gene copy number did not show significant differences between analyzed groups and this gene could be used to normalize the results.

However, ultimately we decided to use the absolute quantification method and the results were calculated based on a standard curve as previously described by Strzałka-Mrozik et al. (2010). This method is not ideal, but each method used to determine changes in the gene expression has its own advantages and disadvantages as reported by Bustin (2002). It is also worth emphasizing that, in the work of Gola et al. (2017) two methods of gene expression quantification were used (absolute and relative) and it was shown that both methods confirmed the changes in the expression of the analyzed genes. Therefore, in our study ACTB mRNA expression assessment was used as positive control of amplification.

Due to Reviewer suggestion, we added information in the section Materials and Methods.

References:

  1. Strzalka-Mrozik B, Stanik-Walentek A, Kapral M, Kowalczyk M, Adamska J, Gola J, Mazurek U. Differential expression of transforming growth factor-beta isoforms in bullous keratopathy corneas. Mol Vis. 2010; 16:161-6
  2. BustinA. Quantification of mRNA using real-time reverse transcription PCR (RT-PCR): Trends and problems. J. Mol. Endocrinol. 2002, 29, 23–39.
  3. Gola J, Strzałka-Mrozik B, Wieczorek E, Kruszniewska-Rajs C, Adamska J, Gagoś M, Czernel G, Mazurek U. Amphotericin B-copper (II) complex alters transcriptional activity of genes encoding transforming growth factor-beta family members and related proteins in renal cells. Pharmacol. Rep. 2017; 69 (6): 1308-1314. doi: 10.1016/j.pharep.2017.05.011

Comments:

“In Figure 3, the authors could include a legend describing the fold-change of differentially expressed genes.”

Answer:

A heatmap is a graphical representation of data that uses a system of color-coding to represent different values. The fold-change value is not provided here. Based on the color change of the fluorescence signals, we evaluate the change in the expression of the studied genes, where the increase in expression is indicated by a change in the color towards red, and the decrease in the expression by a change towards blue. In our study different color pattern was observed on each of the heat maps, and thus it was confirmed that in both study groups (G2 versus G3/G4) the gene expression profiles differed.

Thanks to the Reviewer's suggestion, Figure 1 in the results section has been modified.

Comments:

“Please avoid citing scientific papers published in other languages than English (see citation 11).”

Answer:

The cited scientific paper has been changed to article in English. We apologize for this situation.

Reviewer 2 Report

In this manuscript, the authors performed transcriptomic analyses on tumor samples from 43 patients with Grade2-4 astrocytic brain tumors. They compared expression of TGFb1, b2 and b3 in Grade 2 vs Grade 3-4 tumors. From the rest of the transcriptomic data, they highlighted some pathway involving TGFb and proposed to used detection of TGFb2 mRNA as a biomarker for completing the diagnostic process of astrocytic tumors.

My opinion is that this manuscript needs to be ameliorated to be published for the reason below:

  • The introduction and Discussion need to be reinforced, some publications with the same objective were omitted (ex: Mabrouk et al., Clin. Biol. 2007) and the authors forgot to discuss the physiologic role of TGFb in normal astrocytic cells. Before to discuss the potential role of TGFb from astrocytic tumor, it is important to remind that TGFb has a role in normal astrocytic cells and shows anti-inflammatory activity in this context (ex: Cekanaviciute et al., J Immunol. 2014).
  • The authors mentioned several previous studies including one with 135 tumor samples that drove to similar observation and conclusion. Therefore, it is essential to emphasize the supplementary/novel/original information brought by their own study.
  • In Results, the way the authors normalized values is unclear (page 6,line 5). The way the authors selected the 16 probes (page 8, line 23) is unclear as well: were the only probes with FC>10 ? or were 16 probes selected from a larger panel given FC> 10 ?
  • Concerning the “16” probes, there are indeed 13 probes listed in Table 4. From this table, authors discussed only the few proteins already mentioned in previous publications. If it is important to do that to “validate” the study, it is at least as important to discussed the other proteins because that constitutes the original part of this work.
  • Based on initial observation, if TGFb2 could be distinguished from the other isoforms, why the authors did not try to discriminate in Table 5 signal pathways related more specifically to TGFb2 ? May be that could lead to mechanistic hypothesis to be linked with Grade2 to Grade 4 progression. It is particularly frustrating to show in Table 4 and 5 so many information without any attempt of mechanistic explanation. The STRING analysis should be not restricted to a graph without mechanistic proposal.
  • In Discussion, authors should discuss more in detail their proposal to use TGFb2 as biomarker. Based on their data, how it is possible ? What could be the method ? What could be the cut-off for discriminating between Grade 2 and Grade3-4 ? Even, with significant data, adaptation to a diagnostic test does not appear technically obvious.

Minor point:

Figure 1: to be complete, TGFb3 should be presented as well

Figure 2: to be easier to read, it should be better to present data as in Figure 1, I.e. TGFb1, 2, then 3.

Author Response

Comments:

“I don't feel qualified to judge about the English language and style”

Answer:

The manuscript has been checked and revised by American English Editor. We used Scribendi (online Web-based professional copy editing company). Editor code – EM2189.

Comments:

Does the introduction provide sufficient background and include all relevant references?

Must be improved (x)

Is the research design appropriate? Must be improved (x)

Are the methods adequately described? Must be improved (x)

Are the results clearly presented? Can be improved (x)

Are the conclusions supported by the results? Must be improved (x)”

Answer:

Thankfully to Reviewer’s suggestion the indicated sections were supplemented and improved.

Comments:

The introduction and Discussion need to be reinforced, some publications with the same objective were omitted (ex: Mabrouk et al., Clin. Biol. 2007) and the authors forgot to discuss the physiologic role of TGFb in normal astrocytic cells. Before to discuss the potential role of TGFb from astrocytic tumor, it is important to remind that TGFb has a role in normal astrocytic cells and shows anti-inflammatory activity in this context (ex: Cekanaviciute et al., J Immunol. 2014).”

Answer:

The Introduction and Discussion were reinforced. Additionally, these sections have been supplemented and modified as suggested by the Reviewer.

Comments:

“The authors mentioned several previous studies including one with 135 tumor samples that drove to similar observation and conclusion. Therefore, it is essential to emphasize the supplementary/novel/original information brought by their own study.”

Answer:

We agree with the Reviewer that it is important to highlight original information brought by our own study; therefore appropriate comments have been added to the Discussion section.

Indeed, the study by Zhang et al. (2011) drove to similar conclusions as ours, but they only undertook the assessment of TGFβ1 expression. In our study, we examined the expression of all three isoforms of this cytokine. Furthermore, in the aforementioned study, the authors used an assay exclusively at the protein level. In our study we using oligonucleotide microarrays method and validating the results using RT-qPCR technique, as well as determining the gene expression profile, and performing bioinformatic analysis. The same is true for the other studies cited in the discussion. We did not find any articles describing studies performed on tissues obtained from patients at different grades of astrocytic brain tumor, which comprehensively described the expression of the three TGFβ isoforms and which performed assessment of gene expression profiles and their relationships with TGFβ isoforms using bioinformatic tools. We believe that our study is relevant because of delineation of a TGFβ2 isoform whose expression is statistically significantly different between grades of tumor, and due to the determination of the gene expression profile and the presentation of the network of relationships between genes along with the identification of processes that are associated with tumorigenesis and may influence the progression and malignancy of astrocytic brain tumor. Once again, we would like to thank the Reviewer for the opportunity to highlight the value of our research.

Comments:

“In Results, the way the authors normalized values is unclear (page 6, line 5). The way the authors selected the 16 probes (page 8, line 23) is unclear as well: were the only probes with FC>10 ? or were 16 probes selected from a larger panel given FC> 10 ?”

Answer:

The microarray results were analyzed in the GeneSpring software, where the data was loaded as CEL files and automatically normalized in the software. According to the information contained in the software: '' Raw '' signal values refer to the linear data after summarization using a summarization algorithm (RMA, PLIER, GCRMA, LiWon and MAS5) and normalized signal values are generated after log transformation and baseline transformation in GeneSpring software.

We apologize for unclear explanation of the way we selected the 16 probes (page 8, line 23). Mentioned sentence in Results section was changed to: “After the differentiation criterion was narrowed down to FC > 10.0, from 1,402 probes only 16 characterized by the greatest variation in expression level and their corresponding genes and direction of expression change were selected. “

Comments:

“Concerning the “16” probes, there are indeed 13 probes listed in Table 4. From this table, authors discussed only the few proteins already mentioned in previous publications. If it is important to do that to “validate” the study, it is at least as important to discuss the other proteins because that constitutes the original part of this work.”

Answer:

We apologize for the inconvenience caused by the accidental omission of three probes. We have added the missing elements to the Table 4.

We also thank you for your suggestion regarding the inclusion of a description of the proteins mentioned in the Table 4. Previously, we did not discuss all the proteins listed in the Table 4 in the discussion because we could not find such research describing their expression in tissues obtained from patients with astrocytic brain tumor of various grades, only a minority of them being described in patients without considering the WHO classification. Simultaneously, we felt that studies conducted on cell lines or meta-analyses were not close enough to be comparable. Following your accurate comment about the need to refer to all proteins we have discussed the mentioned proteins based also on data generated by other researchers using bioinformatic methods and obtained in other tumors tissues.

Indeed, the inclusion of the description of these data in the Discussion section was important and significantly increased the value of our study.

Comments:

“Based on initial observation, if TGFb2 could be distinguished from the other isoforms, why the authors did not try to discriminate in Table 5 signal pathways related more specifically to TGFb2 ? May be that could lead to mechanistic hypothesis to be linked with Grade2 to Grade 4 progression. It is particularly frustrating to show in Table 4 and 5 so many information without any attempt of mechanistic explanation. The STRING analysis should be not restricted to a graph without mechanistic proposal.”

Answer:

We would like to thank Reviewer for suggesting that we should try to discriminate in Table 5 signal pathways related more specifically to TGFβ2. For this reason, we have modified Table 5 to emphasize and expose those signaling pathways that are related to the TGFβ2 isoform. However, the table includes selected biological processes and signaling pathways associated with tumorigenesis in which specific differential genes play important roles. These genes include those associated with TGFβ isoforms and genes showing altered expression in gliomas of different grades listed in Table 4.

It should also be remembered that the biological activity of TGFs depends on the quantitative relationships between the individual isoforms of TGFβ1, TGFβ2 and TGFβ3.

Due to the relationships at the molecular level between all the TGFβ isoforms assayed, we decided to include other examples to convey to the readers how genes showing altered expression in astrocytic tumors of different grades are related to processes and signaling pathways potentially associated with tumorigenesis.

Thankfully to Reviewer’s pertinent comment, we have made an attempt of mechanistic explanation in Discussion section.

Comments:

“In Discussion, authors should discuss more in detail their proposal to use TGFb2 as biomarker. Based on their data, how it is possible? What could be the method? What could be the cut-off for discriminating between Grade 2 and Grade3-4? Even, with significant data, adaptation to a diagnostic test does not appear technically obvious.”

Answer:

It should be noted that our basic science study is a preliminary research, which aimed to evaluate the differences in TGFβ isoform expression and changes in gene expression profile depending on the grade of astrocytic brain tumor.

In our conclusions, we just wanted to suggest that statistically significant changes in TGFβ2 expression may in the future serve to improve the diagnosis of this tumor. Currently, the WHO does not recommend specific molecular biology methods to diagnose tumor type in most cases.

In our opinion, in the future, before making a diagnosis, doctors should have access to, for example, determining the level of TGFβ2. Especially since the RT-qPCR method is widely used in the diagnosis of other diseases. Obviously, this method should be validated and approved for diagnostic use. To this end, further studies are necessary, in particular to determine the cut-off value. Due to the number of samples at this point in the research, we were not able to determine the cut-off value, but we intend to continue our work in the future, particularly focusing on the TGFβ2 isoform we selected, in order to distinguish new molecular markers to support the diagnosis of astrocytic brain tumors.

We raised this point in the Discussion section and also modified our conclusions.

Comments:

“Figure 1: to be complete, TGFb3 should be presented as well.”

Answer:

Figure 1 has been completed as suggested by the Reviewer.

Comments:

“Figure 2: to be easier to read, it should be better to present data as in Figure 1, I.e. TGFb1, 2, then 3.”

Answer:

The Figure 2 in the Results section has been improved. The results for each group were presented in different colors. Additionally, the font size in most figures has been increased.

We hope that this makes them much easier to distinguish.

Round 2

Reviewer 2 Report

The authors significantly improved the manuscript by competing Introduction and Discussion. Presentation and explanation of results are were modified in order to present now  a better form to be published.

Study could be still enriched  by mechanistic hypothesis based on data from Table 4 and string analysis in Figure 5, nevertheless, one can consider that a confirmation of such study with more cases is needed before to go to far in mechanistic explanation.